# Enzymatic Regulation of the Gut Microbiota: Mechanisms and Implications for Host Health

**DOI:** 10.3390/biom14121638

**Published:** 2024-12-20

**Authors:** Zipeng Jiang, Liang Mei, Yuqi Li, Yuguang Guo, Bo Yang, Zhiyi Huang, Yangyuan Li

**Affiliations:** 1Guangdong VTR Bio-Tech Co,. Ltd., Zhuhai 519060, China; 2School of Biology and Biological Engineering, South China University of Technology, Guangzhou 510641, China

**Keywords:** enzyme, gut microbiota, regulation, mechanisms, applications

## Abstract

The gut microbiota, a complex ecosystem, is vital to host health as it aids digestion, modulates the immune system, influences metabolism, and interacts with the brain-gut axis. Various factors influence the composition of this microbiota. Enzymes, as essential catalysts, actively participate in biochemical reactions that have an impact on the gut microbial community, affecting both the microorganisms and the gut environment. Enzymes play an important role in the regulation of the intestinal microbiota, but the interactions between enzymes and microbial communities, as well as the precise mechanisms of enzymes, remain a challenge in scientific research. Enzymes serve both traditional nutritional functions, such as the breakdown of complex substrates into absorbable small molecules, and non-nutritional roles, which encompass antibacterial function, immunomodulation, intestinal health maintenance, and stress reduction, among others. This study categorizes enzymes according to their source and explores the mechanistic principles by which enzymes drive gut microbial activity, including the promotion of microbial proliferation, the direct elimination of harmful microbes, the modulation of bacterial interaction networks, and the reduction in immune stress. A systematic understanding of enzymes in regulating the gut microbiota and the study of their associated molecular mechanisms will facilitate the application of enzymes to precisely regulate the gut microbiota in the future and suggest new therapeutic strategies and dietary recommendations. In conclusion, this review provides a comprehensive overview of the role of enzymes in modulating the gut microbiota. It explores the underlying molecular and cellular mechanisms and discusses the potential applications of enzyme-mediated microbiota regulation for host gut health.

## 1. Introduction

The gut microbial community, also known as the gut microbiota or intestinal microbiome, is a vast and diverse population of microorganisms that live in the digestive tracts of humans and other animals. It acts as the first line of defense for the gastrointestinal system. Primarily, the gut microbiota comprises two major phyla, namely Bacteroidetes and Firmicutes, which together account for over 90% of the bacterial population [1]. Other less abundant but equally important phyla such as Actinobacteria, Proteobacteria, Verrucomicrobia, Fusobacteria, and Cyanobacteria contribute to the overall gut microbial community [2].

Despite sharing fundamental roles such as aiding digestion, modulating the immune system [3], maintaining homeostasis [4], influencing metabolism [5], and interacting with the brain–gut axis [6], the composition and functionality of the gut microbiota exhibit distinct variations across species because of factors like diet, genetics, habitat, and lifestyle [7]. Specifically, diet emerges as a pivotal determinant of the gut microbiota community structure and function [8]. One aspect of this is the concept of “diet-dependent diversity”. Many gut bacteria can break down indigestible complex carbohydrates and fibers, producing short-chain fatty acids (SCFAs) such as butyrate, propionate, and acetate. These SCFAs serve as energy sources for colonocytes and influence host metabolism. This means that different dietary inputs result in different metabolic outputs. There are various methods to influence the gut microbial community, both directly and indirectly.

Enzymes from host or microbiota play a pivotal role in modulating the gut microbiota by actively taking part in biochemical reactions that influence the composition, activity, and overall balance of the microbial community [9]. These actions are multifaceted, affecting not only the microorganisms directly but also the gut environment, which shapes the microbial ecosystem [10]. As indispensable catalysts within biological systems, they are involved in the metabolic processes of both the host and the microbes. The role of enzymes in modulating the gut microbiota is multifaceted and significant, including digestive support [11], competitive interactions [12], microbial cross-feeding [13], toxin degradation [14], immune modulation [15], microbial metabolism regulation [16] and reshaping microbial communities [17]. However, understanding the intricate relationship between enzymes and microbial communities, along with the precise mechanisms of enzyme-mediated regulation, remains a significant challenge in scientific research. While studies have shown that specific enzymes can markedly affect the growth and metabolism of certain microbes, the vast diversity of enzymes and their varied impacts on microbial communities complicate the generalization of enzyme-mediated microbial regulation mechanisms.

This article aims to provide an in-depth overview of the role enzymes play in modulating the gut microbiota. It investigates the underlying molecular and cellular mechanisms and explores the potential applications of enzyme-mediated microbiota regulation for improving host gut health. The review presents a holistic view of the complex interplay between enzymes and the gut microbiota, highlighting how we can leverage enzymes to optimize gut microbiota balance and enhance gut health.

## 2. Nutritional and Non-Nutritional Functions of Enzymes and Their Effects on Microbiota

Enzymes have been recognized for their importance for a long time. The definition of an enzyme encompasses three key characteristics: (1) they are synthesized by living cells; (2) they exhibit high efficiency and specificity in their catalytic activity; and (3) they are either proteins or RNA. Their classification is typically based on function and structure, as depicted in Figure 1. Enzymes are classified into seven main categories: oxidoreductases (EC 1), transferases (EC 2), hydrolases (EC 3), lyases (EC 4), isomerases (EC 5), ligases (EC 6), and translocases (EC 7) [18]. In this study, enzymes were categorized into nutritive and non-nutritive groups based on their effects on microorganisms. We also divided the enzymes according to their sources, such as microbial sources, host sources, and exogenous additions.

### 2.1. Nutritional Functions of Enzymes and Their Effects on Microbiota

Enzymes such as proteases, carbohydrases, and lipases are crucial for collapsing complex proteins, carbohydrates, and lipids into absorbable forms [19]. The most representative group of these is carbohydrate-active enzymes (CAZymes), which are specialized in the degradation and synthesis of complex carbohydrates, facilitating the utilization of dietary fibers, including glycoside hydrolases, polysaccharide lyases, glycosyltransferases and non-starch polysaccharide (NSP) enzymes [20]. All these enzymes handle the degradation and synthesis of complex compounds in the gut and are critical for the host to absorb nutrients from the diet, and there are interactions with gut microbes. We present the nutritive functions of enzymes depending on their different sources.

#### 2.1.1. Enzymes Derived from the Microbiota

The vast majority of enzymes generated by microorganisms such as bacteria and fungi play crucial roles in regulating the gut microbiota. These enzymes assist in breaking down complex molecules and enabling the host to digest and absorb nutrients, and they participate in a range of metabolic processes [21]. The presence of microbial enzymes in the gut can improve the efficiency of food digestion, especially for compounds that the host cannot effectively metabolize on its own, such as certain fibers or resistant starches. Certain bacteria can produce cellulase to break down cellulose in plant cell walls, an enzyme that the human body does not produce on its own [22]. Other enzymes help metabolize specific types of sugars like lactose and xylan or are involved in the production of short-chain fatty acids, which are critical for maintaining gut health [23]. Moreover, microbial-generated enzymes can influence the pH within the gut, thereby promoting or inhibiting the growth of other microorganisms indirectly and thus regulating the community structure of the microbiome [24]. In summary, although the host is able to synthesize and secrete a wide range of digestive enzymes, the gut microbiota provides additional enzymatic capacity that complements the host’s digestive processes, ensuring a complete breakdown of dietary components and maximizing nutrient absorption.

#### 2.1.2. Enzymes Derived from the Host

Many digestive enzymes can be secreted and synthesized by the host organism itself, playing essential roles in breaking down food components into absorbable units. These include CAZymes, lipases, proteases, pancreatic enzymes, etc. [25].

CAZymes (carbohydrate-active enzymes): This broad class of enzymes includes glycoside hydrolases, polysaccharide lyases, carbohydrate esterases, and more. They are involved in the breakdown of complex carbohydrates such as starches, cellulose, and hemicellulose into simpler sugars. Lipases are primarily produced in the pancreas and break down dietary fats (triglycerides) into monoglycerides and free fatty acids, which can then be absorbed through the intestinal wall. Proteases, also known as peptidases or proteinases, are a group of enzymes that break down proteins into smaller peptides and amino acids. They are secreted by the stomach (pepsin), pancreas (trypsin, chymotrypsin), and small intestine (carboxypeptidases). Regarding pancreatic enzymes, the pancreas secretes a variety of enzymes that aid in digestion, including those mentioned above (lipases and proteases), as well as amylase, which helps to further digest carbohydrates into simple sugars. These enzymes are part of the host’s intrinsic ability to process nutrients from food. Young animals often experience a delay in the maturation of their gut microbiota because of incomplete secretion of digestive enzymes [26]. This also emphasizes the critical role of host-produced enzymes in animal nutrition, as they break down nutrients into smaller molecules that are more easily digested, absorbed, and utilized, as well as promote the maturation of the microbiota.

#### 2.1.3. Microbial Enzymes Interact with Microbial Communities and Hosts

Interestingly, the microbiota also influences the expression of enzymes, particularly CAZymes. These enzymes, produced by gut microbes, serve various functions within the gut ecosystem. They are involved in breaking down dietary components and modulating other microbial populations. Kaoutari et al. reported that the large diversity of complex polysaccharides in our diet is primarily digested by specialized enzymes encoded by the gut microbiome [27]. A study reported that the protein hydrolysis and starch degradation abilities of bacteria screened after feeding diets with different protein and fiber ratios were different [28]. These findings underscore the symbiotic relationship between enzymes and gut microbes. Enzymes facilitate the synthesis of essential compounds by gut microbes, enhancing their proliferation and metabolic activities. Enzymes produced by gut bacteria can modify bile acids (BAs), which are crucial for the digestion and absorption of dietary fats. This modification also influences the composition of the gut microbiota, highlighting the enzymes’ significant role in gut ecosystem dynamics [29]. Bile acids (BAs) can also serve as modulators of enzyme activity and stability. For instance, they can deactivate prolyl endopeptidases (PEPs) [30]. A study reported the formation of secondary allo-bile acids by novel enzymes from gut Firmicutes [31]. BA-altering enzymes produced by *Bacteroides thetaiotaomicron* have been shown to impact the fitness and metabolism of the gut microbiota, particularly carbohydrate metabolism, by affecting many polysaccharide utilization loci [32]. Similarly, gut bacteria play a crucial role in the synthesis of essential vitamins through their enzymatic activities. For instance, certain vitamins, such as B, E, and K, are produced by the gut microbiota, which are vital for the host’s metabolic processes [33,34,35].

#### 2.1.4. Enzymes Derived from External Factors

Enzymes can work with probiotics to ferment non-digestible substances, offering one way to influence the microbiota. For instance, non-starch polysaccharide (NSP) enzymes, including cellulases, β-glucosidase, xylanase, mannanase, pectinase, and arabinofuranosidase, can convert NSPs into fermentable substrates that support the growth of beneficial bacteria, which play a key role in host metabolism. Research has shown that adding xylo-oligosaccharides (XOSs), fructo-oligosaccharides (FOSs), mannan-oligosaccharides (MOSs), and galacto-oligosaccharides (GOSs) to animal feed can improve gut microbiome populations [36]. Oligosaccharides are produced through the action of NSP-degrading enzymes. Feeding a combination of probiotics with these oligosaccharides has shown synergistic effects. This provides evidence that the products of enzymatic degradation can promote the proliferation of probiotics [37]. Bacteroides, a dominant colonizer in the gut, ferments monosaccharides derived from enzymatic degradation into metabolites such as SCFAs and amino acids [38]. These SCFAs support gut health and serve as an energy source for the host [39]. Some enzymes from food sources are called diet-derived enzymes and also play an important role in the digestion and absorption of nutrients. For example, bromelain is found in pineapple and is a mixture of enzymes that aid in protein digestion. Similarly, papain is present in papaya and aids in protein digestion. These enzymes have anti-inflammatory properties and also exhibit antiarthritic effects and protect against inflammatory conditions [40,41]. Nattokinase is a traditional Japanese food, which is derived from natto. Nattokinase exhibits fibrinolytic and antithrombotic properties, making it a valuable enzyme for cardiovascular health [42]. Several other studies have confirmed the effect of exogenously added enzymes. Kiarie et al. fed poultry and pigs with CAZymes and verified that the enzymes can modulate the gastrointestinal ecology [43]. Hiroki et al. highlighted that the oral supplementation of digestive enzymes alters the composition of the gut microbiota, and *Akkermansia muciniphila* and *Lactobacillus reuteri*, in particular, are over-represented [44].

### 2.2. Non-Nutritional Functions of Enzymes and Their Effects on Microbiota

Enzymes not only play pivotal roles in the digestion and absorption of nutrients but also exhibit a range of non-nutritional functions that significantly impact the gut microbiota. Beyond their traditional role in breaking down macromolecules into absorbable units, enzymes are involved in modulating immune responses, maintaining gut barrier integrity, and influencing microbial composition and activity. Certain enzymes can directly interact with bacterial cell walls or viral particles, altering pathogen viability and virulence. Additionally, enzymes such as defensins and other antimicrobial peptides produced by the host contribute to the regulation of microbial communities, preventing the overgrowth of potentially harmful species. Enzymatic activities within the gut environment also affect the production of short-chain fatty acids (SCFAs) and other metabolites that serve as signaling molecules for host–microbe communication. Moreover, enzymes can influence the pH levels within the gut, thereby shaping the conditions that favor beneficial microbes over pathogens. Figure 1 illustrates a description of the non-nutritive functions of enzymes, such as antibacterial properties and inhibition of pathogens, immune modulation, maintenance of intestinal microbiota balance and gut health, and antioxidative stress. This section explores the diverse non-nutritional roles of enzymes and how these functions affect the intricate balance of the gut microbiota, ultimately impacting overall host health.

#### 2.2.1. Enzymes Derived from the Microbiota

Glucose oxidase (GOD) is an enzyme that can be produced by a variety of sources, including microorganisms such as fungi. Glucose oxidase consumes oxygen and catalyzes β-D-glucose to D-glucono-δ-lactone and hydrogen peroxide (H_2_O_2_), which creates a relatively anaerobic environment in the gut and has a direct effect on the gut microbes [45]. Some enzymes are also involved in interactions with the immune system, contributing to the establishment of a healthy gut environment and preventing pathogen invasion [46]. Liu et al. reported that GOD can improve growth performance, immune function, and intestinal barrier in ducks infected with *Escherichia coli* (*E. coli*) O88, and change the gut microbial community [47]. Other research found that GOD supplementation promotes gut barrier integrity and ileal microbiota balance [48].

Mucin-degrading enzymes are produced by certain bacteria within the gut microbiota. Specifically, Bacteroides and *Akkermansia muciniphila* have been recognized as main mucin degraders. Mucin-degrading enzymes have a dual role: They can collapse the protective mucus layer that lines the gut, which may affect the barrier function and the interaction between the microbiota and the host epithelium [49]. While excessive degradation can weaken the gut barrier, controlled degradation by host and commensal bacteria ensures proper mucus layer turnover [50]. This process allows for nutrient absorption and waste removal while preventing pathogen adhesion. When the mucus layer is maintained at normal levels, beneficial microbes can occupy ecological niches, influencing the proliferation and colonization of pathogens and thus promoting gut health.

#### 2.2.2. Enzymes Derived from the Host

These are enzymes produced by the host and play a crucial role in various physiological processes, including metabolism, inflammation response, and immune function. Sulfotransferases are a family of enzymes that are part of the host metabolism and are involved in the conjugation of compounds with sulfate. They play a role in the detoxification process by increasing the water solubility of compounds and facilitating their excretion [51]. Cyclooxygenase (COX) and lipoxygenase (LOX) are produced endogenously and play a role in the body’s inflammatory response. COX and LOX are involved in producing prostaglandins and leukotrienes, which are important mediators in inflammation [52].

Certain enzymes can selectively target and modulate the growth of specific microbial species, influencing the balance of the gut microbiota. This selective action is crucial for maintaining a healthy gut ecosystem. Lysozyme exemplifies this selectivity, and it also comes from the host, such as human gastric secretions and hen egg whites. A study reported that lysozyme-like proteins secreted by *Bifidobacterium longum* can regulate the human gut microbiota, significantly delaying the growth of some bacteria [53]. Other studies revealed that lysozyme attenuates the dextran sulfate sodium (DSS)-induced colitis by modulating the gut microbiota, increasing the relative abundance of *Akkermansia muciniphila* [54]. Lysozyme enhances the diversity of the gut microbiota by promoting the growth of probiotics and inhibiting pathogens. This action helps to alleviate colon damage and mucosal inflammation in mice with colitis, indicating that lysozyme plays a significant role in modulating the microbial community [55]. Yu et al. demonstrated that the intestinal luminal lysozyme determines the abundance of mucolytic commensal bacteria [56]. Intestinal lysozyme liberates Nod1 ligands from bacteria to direct insulin trafficking in pancreatic beta cells, regulating the microbiota composition close to epithelial cells [57]. Similarly, Paneth cells produce antimicrobial proteins such as lysozyme and cytokines to interact with the gut microbiota mucosal immune system [58]. For example, Paneth cells limit pathogens’ invasion by secreting antimicrobial proteins including lysozyme [59].

#### 2.2.3. Microbial Enzymes Interact with Microbial Communities and Hosts

Enzymes play an important role in regulating intestinal immunity by interacting with gut microbes and the host. They contribute to the breakdown of immunoglobulins and antigens, influencing immune tolerance to the gut microbiota and food antigens. When nutrients are not broken down sufficiently, or when anti-nutritional factor components are present, they can cause damage to the intestinal epithelium cell [60]. Meanwhile, enzymes regulate the acid–base balance, produce prebiotics to probiotics generate volatile fatty acids, and catalyze substances to reduce the intestinal pH, which is also directly related to the gut immune system [61].

Enzymes can modulate the immune system by influencing the production of cytokines and chemokines and affecting the production of antimicrobial peptides or by modulating the activity of immune cells in the gut, which is crucial for gut health and defense against pathogens [62,63]. Experiments have shown that infection pathogens or inflammation in the small intestine often result in reducing disaccharidase enzyme levels and exerting significant effects on the secretion and expression of cellular inflammatory factors [64]. Dual oxidase enhanced the mucosal epithelia function armed its surfaces with efficient microbial control systems and improved the immune function in *Drosophila* [65]. Moreover, enzymes like catalase (CAT) or superoxide dismutase (SOD) can regulate the redox balance of the organism and participate in intestinal immune responses [66]. In inflammatory bowel disease model, *Lactobacilli* with SOD or CAT activity alleviated the inflammation [67].

#### 2.2.4. Enzymes Derived from External Factors

Certain external substances, like environmental toxins and mycotoxins, can affect the host’s health and the composition of the gut microbiota. Enzymes play a crucial role in transforming these potentially harmful substances into less harmful forms that can be excreted by the host. Peroxidase enzymes, in particular, have been shown to catalyze the oxidation of a broad spectrum of environmental pollutants and toxins [68]. Peroxidase enzymes are widely distributed in nature and can be produced from a variety of sources, including plants, animals, and microorganisms, and can also be externally supplemented. Mycotoxins can have detrimental effects on the host, including disrupting intestinal barrier function, interfering with the gut microbiota, affecting nutrient absorption, and triggering intestinal inflammation. However, the use of enzymes can help eliminate these harmful effects by destroying the structure of mycotoxins [69]. Enzymes indirectly protect the stability of the gut microbiota and maintain gut health. Kanwar et al. reported that a dietary supplement containing gut enzymes improved both gut and immune health [70]. Wu et al. used GOD as an additive in poultry feed, finding significant changes in abundance in the phylum Firmicutes, the families Ruminococcaceae and Rikenellaceae, and the genus *Faecalibacterium* [71]. Enzymes produced by the host’s cells or the microbiota contribute to gut health. For instance, intestinal brush border enzymes can directly interact with the gut microbiota, influencing its activity [72].

Understanding these classifications offers a framework for appreciating how enzymes contribute to the complex network of microbial interactions within the gut. This knowledge is essential for shaping the overall health and function of the gut microbiome and for developing potential therapeutic strategies that target the gut microbiota to address various conditions.

## 3. Mechanisms of Enzyme-Mediated Microbial Regulation

Table 1 provides evidence confirming that enzymes can regulate the gut microbiota. A critical aspect of how enzymes regulate microbial communities revolves around substrate availability and competition. Enzymes, whether exogenously added or released by gut bacteria, facilitate the breakdown of complex macromolecules into simpler compounds that can be absorbed by the microbiota. For example, xylanases can break down dietary xylan into oligosaccharides that support the growth of certain gut microbes, such as *Bacteroides*, *Bifidobacterium*, and *Lactobacillus* [73]. The latest research found that cross-feeding and co-culturing are also vital for balanced gut microbiota. The breakdown of complex carbohydrates by Bacteroidetes releases monosaccharides that can be consumed by Firmicutes [74], illustrating how enzyme activity supports a complex web of microbial interactions. Enzymatic products of one microbe can serve as substrates for another bacteria [75]. This interdependency encourages cooperation among various microbial species, enhancing microbial diversity and stability. The synthesis of SCFAs, vitamins, and other metabolites by gut bacteria is often enzyme-dependent, affecting not only the bacteria’s metabolism but also the host’s health and other cross-feeding microorganisms [76]. Changes in substrate availability due to enzymatic activities can shift microbial metabolism toward more energetically favorable pathways. An example is the switch from respiration to fermentation when oxygen becomes limited [77].

Some enzymes, such as GOD, lysozyme, and bacteriophage lysins, can directly influence microorganisms. GOD specifically catalyzes the production of gluconic acid (GA) and hydrogen peroxide (H_2_O_2_) by consuming molecular oxygen. This process creates an anaerobic environment that directly kills harmful bacteria. Another category can degrade peptidoglycan, a major component of bacterial cell walls [87]. Rangan et al. reported that bacteria can secrete bacterial peptidoglycan hydrolase and enhance the tolerance ability with pathogens [88]. Lysozyme is a protein that exerts its enzymatic activity through the hydrolysis of the β-1,4-glycosidic bonds between N-acetylmuramic acid (NAM) and N-acetylglucosamide (NAG) in the polysaccharide backbone of the peptidoglycans of the Gram-positive bacterial cell wall [89]. Wang et al. verified that low-lysozyme significantly enhanced the inhibitory ability of *Staphylococcus aureus* [78]. The researchers found that glycoside hydrolase endolysin from *Acinetobacter baumannii* had high antibacterial potency [79]. Bacteriophage lysins, which are peptidoglycan (PG) hydrolytic enzymes derived from phages, serve as potent tools against multidrug-resistant (MDR) bacteria [90]. Metaviromics analysis has revealed a glycoside hydrolase endolysin with high specificity for Acinetobacter baumannii, demonstrating efficient lytic and antimicrobial activity [91]. The use of endolysins has facilitated the development of broad-spectrum antimicrobials. For instance, phage lysins could be utilized to prevent or treat bovine mastitis, reducing the inflammatory response and pathogen population [80]. Moreover, phage endolysins can adapt to new phage/host environments by acquiring adaptive mutants, highlighting the remarkable adaptability of the phage lysis system [92]. Enzymatic profiling shows that modified intracellular phage endolysins gradually cleave the cell wall of *Bacillus anthracis* [93].

Enzymes can disrupt microbial networks, such as quorum sensing (QS), which is a communication system used by microbes to coordinate their behavior. Enzymes like acyl-homoserine lactone (AHL) lactonases can degrade QS signal molecules, thus modulating the gut microbiota [81]. Many bacteria utilize QS systems to synchronize the expression of the target gene and to coordinate the biological activities of the local population [94]. Blocking QS mechanisms has proven to be a functional and conventional control of infections, and the external enzymes provide an alternative way of reducing communication in pathogenic bacteria that may lead to the degradation of their signal and loss of pathogenicity [95]. The most common signaling molecules in the QS system are acyl-homoserine lactones (AHLs), and quorum quenching is a strategy employed by some bacteria in which they produce enzymes that cleave these AHLs and can prevent biofilm formation and attenuate virulence [82]. The removal of QS molecules by N-acyl homoserine lactonase can elevate the abundance of Proteobacteria and reduce the pathogenic *Aeromonas hydrophila* in the gut [81]. In one study using *P. aeruginosa* as an experimental model, quorum inhibitor and quenching enzymes were investigated, and the results showed that the combination almost completely blocked the QS system of *P. aeruginosa* [83]. In vitro experiments have similarly shown that the quenching enzymes effectively accelerated nitrite accumulation by regulating ammonia, which has a positive effect on enhancing sensing [96]. Quenching enzymes effectively inhibit *Pseudomonas aeruginosa* proliferation, and a combination of QS inhibitors can suppress multiple QS pathways of *Pseudomonas aeruginosa* [83,96]. Using broad-spectrum lactonase preparation can inhibit the QS system in 28 strains, which indicates that the enzymes exhibit the alternative antimicrobial ability and can interfere with or modulate the behavior of microbial communities [84].

Another significant regulatory mechanism involves enzymes mitigating the immune stress response to stressors within the gut, thereby influencing the composition of the intestinal microbiota. Enzymes reduce the incidence of immune stress by degrading substances that resemble immunogenic agents. β-Mannans can elicit an immune response due to their similarity to pathogen surface antigens. This resemblance can lead to unnecessary energy expenditure and immune reactions, as pattern recognition receptors, such as pathogen-associated molecular patterns (PAMPs), mistake β-mannans for actual pathogens in the intestinal tissues, triggering a false immune response [86]. This response can cause the production of inflammatory immune factors and contribute to microbiota dysbiosis [43]. However, enzymes can break down these pathogenic-like structures, reducing unnecessary intestinal stress and preventing immune system depletion and microbiota imbalances. Enzymatic intervention can improve nutrient absorption and help maintain the balance of the microbiota, contributing to overall intestinal health.

## 4. Application of Enzyme-Mediated Regulation of Gut Microbes

As biocatalysts, enzymes have emerged as key modulators of microbial interactions within the gut. They can directly target microbial components or indirectly influence microbial functions, impacting host health. Table 2 demonstrates the effects of enzymes on gut microbes and overall gut health.

Enzymes have been widely utilized in livestock farming. For instance, Qu et al. used glucose oxidase (GOD) as a feed additive for livestock and discovered that GOD could prevent mycotoxin contamination and rebalance the gut microbiota disrupted by mycotoxins. This is because the products of the GOD-catalyzed process can bind to mycotoxins, neutralizing their toxicity and maintaining microbiota balance [106]. Supplementation with GOD has been found to increase gut bacterial diversity and the abundance of beneficial bacteria in broiler chickens [48,71]. GOD has been reported to have a positive impact on the intestinal health of broilers by specifically enhancing the genera Eubacterium and Christensenella, which are positively correlated with improved intestinal digestive enzymatic activities, growth performance, and meat quality [107]. This enhancement in specific bacterial genera contributes to the overall improvement in gut health, which suggests that GOD not only promotes specific beneficial bacteria but also contributes to a more diverse and balanced gut microbiota, which is crucial for optimal gut health. The prebiotic effects of yeast mannan (YM) and the supplementation of β-mannanase have been evaluated for their potential to promote beneficial Bacteroides species in the gut, and these species improve the intestinal environment, which positively impacts host health by enhancing the proliferation of probiotics and inhibiting harmful bacteria [102]. β-mannanase has been shown to improve immune function and the gut microbiota in broiler chickens and weaned piglets. By fostering a more balanced gut microbiota, these interventions not only support specific beneficial bacteria but also contribute to overall gut health and immune function [103,104].

Enzymes provide a range of therapeutic benefits, including disease treatment, a reduction in oxidative stress, and the regulation of immune responses. The release of SodA from lysed Lactococcus lactis I-1631 in the colonic lumen has been demonstrated to alleviate oxidative stress and reduce colitis symptoms. This enzyme converts superoxide anions into H_2_O_2_, thereby decreasing superoxide levels in colonic epithelial cells [87]. The artificial enzyme FeSA, mimicking the functions of SOD and CAT, has been incorporated into *Bifidobacterium longum* (BL) probiotics for the treatment of inflammatory bowel disease (IBD). This approach has shown promise in modulating the intestinal immune microenvironment and regulating dysbiosis in the gut microbiota, which indicates that the enzymes can modulate the gut microbiota directly [97]. Recent studies have underscored the role of ACE2 in the modulation of the gut microbiota. In COVID-19 patients, SARS-CoV-2 downregulates ACE2, leading to increased intestinal permeability and inflammatory responses, which can disrupt the gut microbiota balance [98]. Alkaline phosphatase (IAP) has demonstrated potential in preventing alcohol-induced intestinal inflammation by reducing levels of pro-inflammatory cytokines such as TNF-α and IL-1β [99]. A novel fusion protein, LHD, combining phage lysin and human α-defensin 5 (HD5), has shown potent bacteriolytic activity against *Clostridioides difficile*, a significant pathogen in intestinal infections. This enzyme not only lyses C. difficile but also inhibits the glycosylation activity of toxin B, reducing its cytotoxicity [100]. DAO and D-amino acids have been found to exhibit antimicrobial effects, particularly against Vibrio species, by producing H_2_O_2_. This enzyme can also modulate gut microbiota composition, promoting probiotics and inhibiting harmful bacteria, thus playing a role in mucosal defense [101]. Using quenching enzymes, such as AiiO-AIO6, has demonstrated the potential to promote fish growth by modulating the gut microbiota, increasing the abundance of beneficial bacteria, and reducing harmful species [105]. One approach involves the use of a quenching quorum enzyme produced by the probiotic *Bacillus* sp. QSI-1, which regulates AHL (N-acyl homoserine lactone) signaling molecules in the fish gut. This method has been shown to significantly reduce Aeromonas spp. in the fish gut [81].

Enzymes play a critical role in shaping the gut microbiota and show promise across various application conditions. However, it is important to note that while the therapeutic potential of enzymes is encouraging, much of the research is still in its early stages. Further extensive clinical studies are required to fully understand their mechanisms of action and explore their potential applications in gut health.

## 5. Conclusions and Future Perspectives

In summary, enzymes play a crucial role in the functioning and regulation of microbial communities. They are not only catalysts but also serve as key regulatory elements. Enzymes significantly shape the structure, function, and adaptation of these communities to their environmental context, highlighting the complex interplay between enzymes and the ecosystems they inhabit.

Enzymes perform a variety of functions, including the breakdown and absorption of nutrients, exhibiting antibacterial properties, and modulating immune responses. These actions influence the composition and activity of microbial communities, impacting overall gut health and host metabolism (Figure 1). Enzymes function through diverse mechanisms: some stimulate the growth of beneficial microbes, others kill harmful pathogens directly, some inhibit quorum sensing, and others enhance the structure and composition of the intestinal microbiota by modulating immune functions (Figure 2). In this study, we focused on the interaction between enzymes and bacteria. Enzymes offer unique benefits and hold significant potential in regulating intestinal microorganisms. The modulation of the gut microbiota through enzymatic intervention presents a promising approach for managing gastrointestinal health. The enzymes discussed in this review have demonstrated varied mechanisms of action, from enhancing beneficial microbial populations to regulating immune responses. Further research is necessary to elucidate the full spectrum of these enzymes’ impacts and to translate these findings into clinical applications.

The future development of enzymes to modulate gut microbes may concentrate on several key areas: (1) enzyme engineering—enhancing enzyme efficacy and stability through protein engineering and directed evolution methods: Notably, enzymes need to be stable and active under the harsh conditions of the stomach, including acids and protein hydrolases that may degrade the enzymes. Moreover, alternative methods of overexpressing certain enzymes should be explored through engineered microorganisms. This approach has the potential to be used as a therapeutic strategy by using engineered microorganisms as living factories that produce and secrete desired enzymes directly in the host’s environment; (2) high-throughput screening—employing high-throughput techniques to swiftly identify enzymes with specific traits and to discover novel enzyme sources via metagenomics and other omics technologies; (3) computational biology—leveraging computational approaches to predict enzyme structures and functions, aiding in enzyme design and optimization, and examining the synergistic effects of enzymes to develop multi-enzyme systems that boost the efficiency of various industrial processes. Addressing current challenges and investing in these research areas will be crucial in unlocking the full potential of enzymes, leading to breakthroughs in health, industry, and environmental sustainability.

## Figures and Tables

**Figure 1 biomolecules-14-01638-f001:**
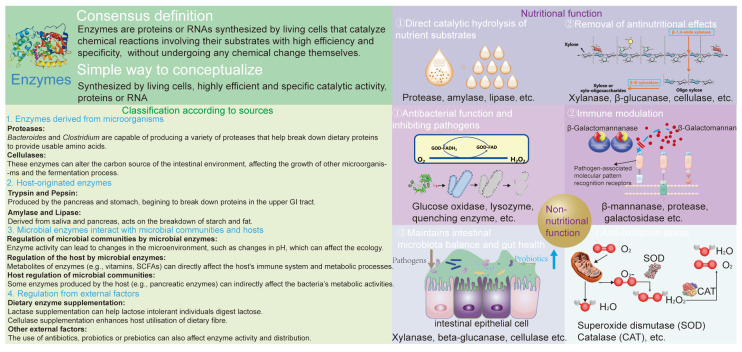
Classification of enzymes and their functions.

**Figure 2 biomolecules-14-01638-f002:**
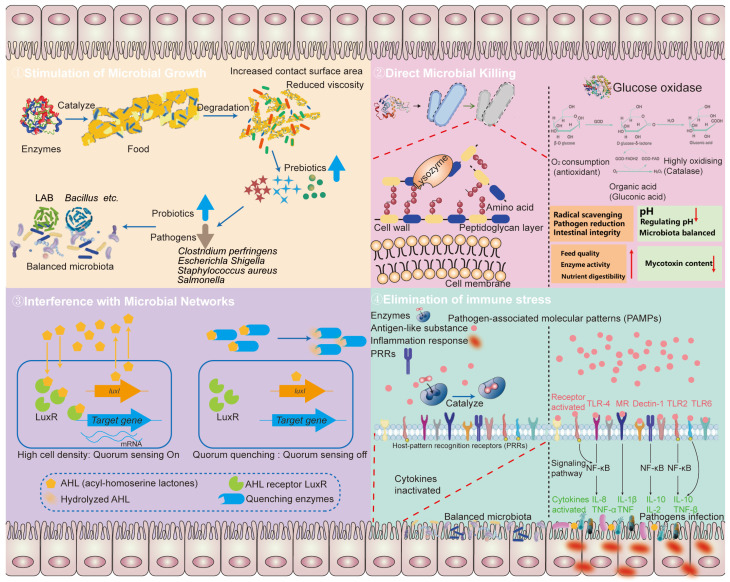
Mechanisms of enzyme regulation of gut microbes. The main ways in which enzymes regulate the intestinal microbiota include (1) stimulation of microbial growth: enzymes can stimulate the growth of beneficial gut microbes; (2) direct microbial killing: certain enzymes can kill gut microbes directly; (3) interference with microbial networks: enzymes can disrupt microbial networks, such as quorum sensing (QS), which is a communication system used by microbes to coordinate their behavior; and (4) alleviating the immune stress: the use of enzymes to reduce the occurrence of immune stress is through the degradation of resemble immunogenic substances.

**Table 1 biomolecules-14-01638-t001:** Mechanisms of enzyme regulation in the gut microbiota.

Function	Type of Enzyme	Origin of the Enzyme	Treatment	Gut Microbiota Influence	Reference
Stimulation of Microbial Growth	Xylanases	Fungi	Breakdown of dietary xylan into oligosaccharides	Promotes the growth of beneficial bacteria such as *Bacteroides*, *Bifidobacterium*, and *Lactobacillus*	[73]
Direct Microbial Killing	Lysozyme	From hen egg white; Paneth cells	Hydrolyzes β-1,4-glycosidic bonds in peptidoglycans of Gram-positive bacteria	Enhances inhibitory ability against *Staphylococcus aureus*	[78]
Direct Microbial Killing	Glycoside Hydrolase Endolysin	Phage AbTZA1	High antibacterial potency against *Acinetobacter baumannii*	Effective against multidrug-resistant (MDR) bacteria	[79]
Direct Microbial Killing	Bacteriophage Lysins	Phage enzymes	Serve as potent tools against multidrug-resistant (MDR) bacteria	Utilized to prevent or treat bovine mastitis, reducing the inflammatory response and pathogen population	[80]
Disruption of Microbial Networks	Acyl-Homoserine Lactone (AHL) Lactonases	*Bacillus* sp.QSI-1	Degrades QS signal molecules	Reduces communication in pathogenic bacteria, leading to decreased pathogenicity	[81]
Disruption of Microbial Networks	Acyl-Homoserine Lactone (AHL) Lactonases	Gram-negative, Gram-positive bacteria and archaea	Cleaves AHLs to prevent biofilm formation and attenuate virulence	Elevates the abundance of *Proteobacteria* and reduces pathogenic *Aeromonas hydrophila* in the gut	[82]
Disruption of Microbial Networks	Quorum Inhibitors and Quenching Enzymes	*Bacillus* species	Combination treatment blocks the QS system of *Pseudomonas aeruginosa*	Effectively inhibits *Pseudomonas aeruginosa* proliferation and suppresses multiple QS pathways	[83]
Disruption of Microbial Networks	Broad-Spectrum Lactonase Preparation	Estuarine bacteria	Inhibits QS system in 28 strains	Exhibits alternative antimicrobial ability and interferes with microbial community behavior	[84]
Mitigation of Immune Stress	Antigen degradation enzymes	Bacteria	Degradation of substances resembling immunogenic agents	Reduces unnecessary intestinal stress and prevents immune system depletion and microbiota imbalances	[85]
Mitigation of Immune Stress	β-Mannanase	Fungi	Breaks down β-mannans, reducing false immune responses	Improves nutrient absorption and helps maintain the balance of the microbiota	[86]

**Table 2 biomolecules-14-01638-t002:** Summary of effects of enzymes on mediating gut microbes.

Type of Enzyme	Origin of the Enzyme	Treatment	Host Health Influence	Reference
GOD	*Penicillium notatum*	A total of 525 one-day-old healthy AA broiler chickens were randomly divided into five groups: a control group, an antibiotic growth promoter (AGP) supplementation group, and three groups supplemented with different concentrations of GOD at 40 U/kg, 50 U/kg, and 60 U/kg, respectively.	The GOD supplementation significantly increased the abundance of *Faecalibacterium prausnitzii*, *Ruminococcaceae*, and Firmicutes in the cecum and decreased the abundance of *Rikenellaceae*. Compared to the AGP group, the GOD-supplemented groups significantly enhanced gut bacterial diversity.	[71]
GOD	*Aspergillus niger*	Supplementation with GOD at a dosage of 150 U/kg was used to prevent and mitigate necrotic enteritis (NE) caused by *Clostridium perfringens* (Cp) in broiler chickens.	Infection with Cp alters the structure of the ileal microbiota, and supplementation with GOD can partially reverse these changes. There was a trend towards an increased relative abundance of *Helicobacter* and a decrease in *Streptococcus* with GOD supplementation.	[48]
Superoxide dismutase (SodA)	*Lactococcus lactis*	The impact on intestinal inflammation was investigated in mouse models treated with DSS and lacking T-bet, Rag-2, or Il-10 genes.	Following bacterial lysis, SodA is released into the colonic lumen. SodA converts superoxide anions (O2-) into H_2_O_2_, reducing oxidative stress. The released SodA reacts with superoxide anions, decreasing superoxide levels in colonic epithelial cells and alleviating symptoms of colitis.	[87]
The artificial enzyme FeSA (mimic SOD and CAT function)	Artificial-enzymes-armed *Bifidobacterium* *longum*	Ulcerative colitis (UC) and Crohn’s disease (CD) models were established in mice and beagle dogs to assess the therapeutic efficacy of BL@B-SA. In the mouse model, UC was induced by administering water containing 3% DSS. In the beagle dog model, UC was induced by the intracolonic injection of 7% acetic acid.	BL@B-SA significantly increased the relative abundance of beneficial bacteria, such as those from the Lachnospiraceae family, while reducing the relative abundance of harmful bacteria, such as those from the Enterobacteriaceae family, thereby regulating dysbiosis in the gut microbiota.	[97]
Angiotensin-converting enzyme 2 (ACE2)	Type II pneumocytes in the lungs,	Retrospective studies and systematic reviews have collected data on gastrointestinal symptoms and viral loads in COVID-19 patients. The role of ACE2 in inflammatory bowel disease (IBD) is highlighted by findings that show a 60% lower expression of ACE2 in inflamed areas of CD patients compared to healthy individuals, yet an increased expression in the colon.	SARS-CoV-2 downregulates ACE2 in the gut via its spike protein, leading to increased intestinal permeability and inflammatory responses. This disruption can destabilize the equilibrium of the gut microbiota, reducing the abundance of beneficial bacterial species such as *Faecalibacterium prausnitzii*, while increasing the levels of harmful species like *Coprobacillus* and *Clostridium ramosum*.	[98]
Alkaline Phosphatase	Hepatocytes (liver)-Osteoblasts (bone)- Enterocytes (intestinal epithelium)	IAP was incorporated into the liquid diet at a dose of 200 U/mL and administered orally to mice for 10 consecutive days.	Pretreatment with IAP significantly reduced the levels of TNF-α and IL-1β in intestinal tissue, indicating its potential to prevent alcohol-induced intestinal inflammation.	[99]
Phage lysozyme	Intestinal brush-border enzyme	A novel phage lysin–human defensin fusion protein (LHD), integrating the functional domains of phage lysin and human α-defensin 5 (HD5), and demonstrating potent bacteriolytic activity.	In a mouse model, administration of the LHD protein significantly alleviated symptoms of *Clostridioides difficile* infection (CDI), reduced mortality, and markedly decreased the number of *C. difficile* spores and toxin levels in feces.	[100]
D-amino acid oxidase (DAO)	Enterocytes of the small intestine	Specific pathogen-free (SPF) and germ-free (GF) mice were selected for the quantification of D-amino acids in cecal contents. Samples of small intestinal epithelial cells from SPF and GF mice were analyzed for DAO expression and functionality.	DAO can modulate the composition of the gut microbiota, increasing the abundance of probiotics such as lactobacilli and decreasing the abundance of harmful bacteria like Bacteroides. The absence of DAO leads to elevated sIgA levels, which affects the balance of the gut microbiota.	[101]
Oligosaccharide degrading enzyme	*Saccharomyces cerevisiae*	The prebiotic effects of YM in humans, as well as its impact on the gut microbiota and skin condition, were also evaluated.	Through the action of enzymes, *Bacteroides thetaiotaomicron* and *Bacteroides ovatus* can use YM as a carbon and energy source, increasing their abundance in the gut improving the intestinal environment, and positively impacting host health.	[102]
β-mannanase	*Bacillus lentus*	The control group (NC) was fed a low-energy diet with a reduction of 50 kcal/kg, while the experimental group (NC+BM) received the same low-energy diet supplemented with 100 mg/kg of β-mannanase.	β-mannanase can promote the proliferation of probiotics such as *Lachnospiraceae* and inhibit the colonization of *Pseudomonas* in the gut. It also suppresses microbial fatty acid degradation by reducing the activity of glutaryl-CoA dehydrogenase.	[103]
β-mannanase	*Bacillus subtilis*	β-mannanase (150 mg/kg for 42 days) was used to enhance the growth performance, intestinal barrier function, and gut microbiota of weaned piglets.	The fecal microbial community structure in the β-mannanase group significantly differed from that of the control group. Specifically, the β-mannanase group had lower proportions of *Desulfobacterota*, *Lachnospiraceae*_*NK4B4*_*group*, and *Chlamydia*, while having a higher proportion of *Paludicola* in their feces.	[104]
Quenching enzyme	*Bacillus* sp. QSI-1	A method was employed to regulate the AHL (N-acyl homoserine lactone) signaling molecules in the fish gut by using the probiotic *Bacillus* QSI-1, which produces a quenching quorum enzyme. These enzymes were mixed with a basic diet to create QSI-1-supplemented feed.	Counts of total bacteria, lactic acid bacteria (LAB), *Bacillus* spp., *E. coli*, and *Aeromonas* spp. in the fish gut revealed a significant reduction in *Aeromonas* spp. in the QSI-1 treatment group.	[81]
Quenching enzyme	*Ochrobactrum* sp. M231	To investigate the impact of the enzyme AiiO-AIO6 on fish growth, focusing on the modulation of the gut microbiota. A control group was fed a basic diet, while the AiiO-AIO6 treatment group was fed the same diet supplemented with AiiO-AIO6 (5 U/g).	At the genus level, the AiiO-AIO6 treatment led to a significant increase in the abundance of *Ralstonia*, *Rhodococcus*, and *Lactobacillus*. Conversely, the relative abundance of bacteria such as *Legionella*, *Pseudorhodoplanes*, and *Gemmobacter* was significantly reduced.	[105]

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
