# Peer review of "Enzymatic Regulation of the Gut Microbiota: Mechanisms and Implications for Host Health"

_biomolecules, 2024, doi:10.3390/biom14121638_

Round 1
Reviewer 1 Report
Comments and Suggestions for Authors
This review by Jiang et al. merits consideration for publication in Biomolecules. It primarily reviews the enzymatic arsenal of the gut microbiota and how it could be used for fields such as medicine and animal agriculture.
However, there are a few points I would like addressed.
Major Points
Figure 1 is too crowded. I understand the various classifications of the enzymes, however, some of them are irrelevant for the scope of this review, such as enzymatic structure. The authors could perhaps divide the enzymes into host- and gut microbiota-derived classes, as a general overview. Also, diet-derived enzymes can be included. The figure should also show the intertwining of the classes of enzymes. The authors should keep their nutritional and non-nutritional classifications, which is an important concept.
To further elaborate on this point, the authors should consider writing a few lines on host-derived and diet-derived enzymes. They do mention lysozyme, which is of mammalian origin as well as bacterial, although they mention viral lysozyme. This is important as not only bacterial enzymes are found in the intestine, as there are many digestive- and diet-derived enzymes that could affect microbiota regulation. I understand this is not a focal point of the review, but there should be a mention of them.
There are some instances of reporting studies without context - the authors should present the findings of these studies and make conclusions relevant to the scope of their review in a comprehensive manner.
For example, lines 313-328, where there is mention of glucose oxidase, the authors state (line 317):
"Compared with antibiotic group, using GOD improved the genera Eubacterium and Christensenella which were positively related to the improved intestinal digestive enzymatic activities, growth performance, and meat quality of broilers[96]. Supplementation with GOD has been found to enhance gut bacterial diversity and increase the abundance of beneficial bacteria in broiler chickens, suggesting its potential in promoting gut health[45, 47]."
There is no clear cohesion in this block of text, one sentence mentions specific bacterial genera and improvement of enzymatic activities, and the next line discusses bacterial diversity and gut health. I understand this is in relation to glucose oxidase, however, I would suggest to the authors to improve the overall coherence of their text. It is important not to just spew out references.
In the conclusion and future perspectives section, the authors should mention that any engineered enzyme will have to bypass the gastric juices of the stomach, which is major hurdle. Another alternative could be to engineer microbes to overexpress certain enzymes and employ these as medicines.
Minor Points
Tables 1 and 2 should include the origin of the enzyme.
Figure 2 is informative, but the font is too small.
Lysozyme is an glycosylase, not a peptide (line 144).
Overall, the nutritional and non-nutritional classifications of microbial enzymes, and effects on gut microbiota regulation, is an important concept reviewed in this manuscript. However, sometimes, the main message is lost.
Comments on the Quality of English Language
I would suggest to the authors to have a native English speaker review their manuscript to improve the coherence.
Author Response
Thank you for your recognition of our work, we have made revisions based on your suggestions which have been very helpful!
Comments 1: Figure 1 is too crowded. I understand the various classifications of the enzymes, however, some of them are irrelevant for the scope of this review, such as enzymatic structure. The authors could perhaps divide the enzymes into host- and gut microbiota-derived classes, as a general overview. Also, diet-derived enzymes can be included. The figure should also show the intertwining of the classes of enzymes. The authors should keep their nutritional and non-nutritional classifications, which is an important concept.
Response 1: Thank you for your suggestions. We had revised the Figure 1, and divided the enzyme into four group: Enzymes derived from the microbiota, Enzyme coming from the host, Microbial enzymes interact with microbial communities and hosts. Besides, we maintained the nutritional and non-nutritional classifications. Modifications reflected in 2.1 and 2.2.
Comments 2: To further elaborate on this point, the authors should consider writing a few lines on host-derived and diet-derived enzymes. They do mention lysozyme, which is of mammalian origin as well as bacterial, although they mention viral lysozyme. This is important as not only bacterial enzymes are found in the intestine, as there are many digestive- and diet-derived enzymes that could affect microbiota regulation. I understand this is not a focal point of the review, but there should be a mention of them.
Response 2: Thank you for your suggestion to include additional information on host-derived and diet-derived enzymes in our review. We agree that this would provide a more comprehensive view of the factors influencing gut microbiota regulation. To address this, we had added a few lines about the role of both diet-derived enzymes and host-derived in the intestinal environment.
Line 172-179, 391-399.
Comments 3: There are some instances of reporting studies without context - the authors should present the findings of these studies and make conclusions relevant to the scope of their review in a comprehensive manner.
For example, lines 313-328, where there is mention of glucose oxidase, the authors state (line 317):
"Compared with antibiotic group, using GOD improved the genera Eubacterium and Christensenella which were positively related to the improved intestinal digestive enzymatic activities, growth performance, and meat quality of broilers[96]. Supplementation with GOD has been found to enhance gut bacterial diversity and increase the abundance of beneficial bacteria in broiler chickens, suggesting its potential in promoting gut health[45, 47]."
There is no clear cohesion in this block of text, one sentence mentions specific bacterial genera and improvement of enzymatic activities, and the next line discusses bacterial diversity and gut health. I understand this is in relation to glucose oxidase, however, I would suggest to the authors to improve the overall coherence of their text. It is important not to just spew out references.
Response 3: Thank you for your insightful comments and suggestions for improvement. We understand your concern regarding the coherence and context of the discussion on glucose oxidase (GOD) in our manuscript. Your feedback is well-taken, and we agree that the presentation of findings should be more integrated and coherent to maintain the flow and relevance of the text. Line 656-675.
Comments 4: In the conclusion and future perspectives section, the authors should mention that any engineered enzyme will have to bypass the gastric juices of the stomach, which is major hurdle. Another alternative could be to engineer microbes to overexpress certain enzymes and employ these as medicines.
Response 4: Thank you for your insightful comment on our manuscript. You are correct in pointing out the importance of considering the challenges that engineered enzymes or microbes would face in the gastric environment. We have taken your suggestions into account and have made the revisions to our conclusion and future perspectives section. Line 764-770.
Comments 5: Tables 1 and 2 should include the origin of the enzyme.
Response 5: We had added the relevant contents in Table 1 and Table 2.
Comments 6: Figure 2 is informative, but the font is too small.
Response 6: Thank you for your comment, and we had revised it.
Comments 7: Lysozyme is an glycosylase, not a peptide (line 144).
Response 7: Thank you for your suggestion. We had made the change. Line 402.
Comments 8: Overall, the nutritional and non-nutritional classifications of microbial enzymes, and effects on gut microbiota regulation, is an important concept reviewed in this manuscript. However, sometimes, the main message is lost.
Response 8: Thank you for your valuable suggestions. We have optimized the structure of the framework while retaining it to ensure that key information is not missed.
Reviewer 2 Report
Comments and Suggestions for Authors
In the present review, "Enzymatic Regulation of Gut Microbiota: Mechanisms and Implications for Host Health" the authors aimed to shell the implication enzymatic pathways on regulation of the gut microbiota and the host, which is an emerging point of view of how the microbiota itself and with the host might interact.
The authors reviewed various publications related to the topic and mentioned in the present manuscript the major of the enzymatic routes recently published, however when reading the manuscript, it is difficult to identify which routes/ enzymes mentioned (from the microbiota/host) came from the microbial compartment, the host or the environment (e.g.: diet supplemented).
The current division between nutritional vs non-nutritional enzymes is quite useful as the focus in this review is the intestine, but on top of that, deciphering both microbial and host compartment is a MUST and should independently the division from:
- Enzymes coming from the microbiota
- Enzymes coming from the host
- How microbial enzymes regulate microbial composition and the host and vice versa.
- Regulation from external factors (e.g.: diet supplemented)
Schematizing organizing in a clear way all the current information already present in the manuscript will significantly improve the review. Authors are already mentioning all these topics but when reading it’s difficult to ascertain which enzymes/pathways are coming from.
Moreover, and from the biochemical point of view, it is surprising that a review focusing on the enzymes, is not using the “official scientific classification” of them (e.g.: https://pubmed.ncbi.nlm.nih.gov/34773359/). This should also be included in the present manuscript.
Other comments:
Line 41 (and others): preventing diseases, in all manuscript authors mention microbiota plays a role in preventing diseases, but some commensals can also cause disease (e.g.: pathobionts), therefore this should be included in all mentions.
Line 158: change panth cells to paneth cells
Line 467: Modify reference format
Authors are using microbiota, this includes fungi, viruses and bacteria. They include yeasts in Fig 2, but no clear mentions of them in the manuscript. If main research in enzymes only comes from bacteria, it should be described in the manuscript or include the other members of the microbiota.
Comments on the Quality of English LanguageEnglish is not bad, but a better organization of the manuscript will help in their undestanding.
Author Response
Thank you for your recognition of our work, we have made revisions based on your suggestions which have been very helpful!
Comments 1: The authors reviewed various publications related to the topic and mentioned in the present manuscript the major of the enzymatic routes recently published, however when reading the manuscript, it is difficult to identify which routes/ enzymes mentioned (from the microbiota/host) came from the microbial compartment, the host or the environment (e.g.: diet supplemented).
The current division between nutritional vs non-nutritional enzymes is quite useful as the focus in this review is the intestine, but on top of that, deciphering both microbial and host compartment is a MUST and should independently the division from:
- Enzymes coming from the microbiota
- Enzymes coming from the host
- How microbial enzymes regulate microbial composition and the host and vice versa.
- Regulation from external factors (e.g.: diet supplemented)
Schematizing organizing in a clear way all the current information already present in the manuscript will significantly improve the review. Authors are already mentioning all these topics but when reading it’s difficult to ascertain which enzymes/pathways are coming from.
Response 1: Thank you for your valuable suggestions. We had made the relevant changes in accordance with your request, mainly reflected in sections 2.1 and 2.2.
Comments 2: Moreover, and from the biochemical point of view, it is surprising that a review focusing on the enzymes, is not using the “official scientific classification” of them (e.g.: https://pubmed.ncbi.nlm.nih.gov/34773359/). This should also be included in the present manuscript.
Response 2: Thank you for your valuable comments. We had added relevant content in the article. Line 85-87.
Comments 3: Line 41 (and others): preventing diseases, in all manuscript authors mention microbiota plays a role in preventing diseases, but some commensals can also cause disease (e.g.: pathobionts), therefore this should be included in all mentions.
Response 3: Thank you for your comments. We had revised this issue and added the side effect of commensals. Line 11 and Line 47.
Comments 4: Line 158: change panth cells to paneth cells
Response 4: We had made the change.
Comments 5: Line 467: Modify reference format
Response 5: We had modified its format.
Comments 6: Authors are using microbiota, this includes fungi, viruses and bacteria. They include yeasts in Fig 2, but no clear mentions of them in the manuscript. If main research in enzymes only comes from bacteria, it should be described in the manuscript or include the other members of the microbiota.
Response 6: Thank you for your suggestion. We focused on bacteria, so we had stated this in the text and modified the content in Figure 2. Line 754-755.
Round 2
Reviewer 1 Report
Comments and Suggestions for Authors
I thank the authors for addressing my concerns. The content of this review paper is now rich and insightful.
However, the authors should polish their sentences by having a native English speaker read their text.
For example, It's not "another research" it's "other research" or "a study" (line 378). Overall the English quality is much improved from the original submission, however, there are few blemishes that can polished out.
Author Response
Comment1: I thank the authors for addressing my concerns. The content of this review paper is now rich and insightful.
However, the authors should polish their sentences by having a native English speaker read their text.
For example, It's not "another research" it's "other research" or "a study" (line 378). Overall the English quality is much improved from the original submission, however, there are few blemishes that can polished out.
Response 1: Thank you for your recognition of the majority content of the manuscript, but there may be some errors in the manuscript, and we had revised it with the supervision of native English Speaker. Besides, the text was replaced with correct language and better descriptions. (The errors such as “another research” had been revised in this manuscript). We really hope that the flow and language level have been substantially improved.
Reviewer 2 Report
Comments and Suggestions for Authors
The authors have adressed the majority of the comments of the first revision, however few points can be further improved:
- line 12 and 13, one of the "shape" might be changed by a synonym
- line 23: this paragraph is not 100% undestandable. The english and the writting need to be improved
- line 47: change preventing disease by maintaining homeostasis for example. Microbiota by default are not preventing diseases.
- line 58: "enzymes" clarify from which source come from them (host, microbiota I guess).
- line 89: clarify what do you mean by source.
- line109: this section must only include enzymes comming from the gut microbiota as you are refering to that in the review.
- line 110: "exogenously" this section is entitled enzymes from the microbiota and you are descriving the gut commensal microbiota, all the enzymes supplemented, independelty if they came or not from bacterial synthesis, must be moved to "regulation of external factors section - line 171" as they are supplementes and then must be considered external.
- line 119: "feeding", same comment as previous. All the enzymes supplemented, independelty if they came or not from bacterial synthesis, must be moved to "regulation of external factors section - line 171" as they are supplementes and then must be considered external.
- line 121: what do you mean by examining the rats eating...." this sentence needs to be improved from a scientif and english point of view.
- line128: this section must only include enzymes comming from the host as you are refering to that in the review.
- line 132: "feed" if enzymes are fed, this need to be moved to "regulation of external factors section - line 171" as they are supplementes and then must be considered external.
- line 165: Bacteroides Thet... write in italics.
- line 171: change title to Enzymes comming from ecternal factors or similar, to include all your observations not comming from the gut microbiota or the host.
- line 180: have in mind for this section all the comments from the last section (2.1).
- line 181: rewrite the introductory paragraph as done in line 100 to maintain homogeineity of the review.
- Table 1, lysozyme also comming from paneth cells no?
- line 623: summary must go to final summary and conclusions section
- Table 2: for ACE2 change Host by the cell types secreting it
- Table 2: for Alkaline Phospatase include the cell types secreting it
- Table 2: for DAO change Host by the cell types secreting it
- Table 2: for ACE2 change Host by the cell types secreting it
- Table 2: for Oligoscharide degradins enzyme change Yeast by the yest specie secreting it
- line 987: equalize reference style
Comments on the Quality of English LanguageEnglish from few parts must be improved to make the review undestandable. See previous section.
Author Response
Comments and Suggestions for Authors
Comment 1: The authors have adressed the majority of the comments of the first revision, however few points can be further improved:
Response 1: We thank the Reviewer for his/her carefully reading. Your careful examination of my thesis and your suggestions have made my work by going up one flight of stairs.
Comment 2: - line 12 and 13, one of the "shape" might be changed by a synonym
Response 2: Thank you for your suggestion. Line 12 and 13, we changed the ‘shape’ by a synonym.
Comment 3: - line 23: this paragraph is not 100% undestandable. The english and the writting need to be improved
Response 3: We have revised this paragraph in the manuscript. In particular, lines 22-25.
Comment 4- line 47: change preventing disease by maintaining homeostasis for example. Microbiota by default are not preventing diseases.
Response 4: Thank you for your suggestion, we changed this sentence. Line 57.
Comment 5: - line 58: "enzymes" clarify from which source come from them (host, microbiota I guess).
Response 5: We truly appreciate the reviewer's attentive reading. We added the relevant contents in this sentence. Line 68.
Comment 6: - line 89: clarify what do you mean by source.
Response: 6: We appreciate your insightful comments on the manuscript. And we revised this sentence. Line 99-101.
Comment 7: - line109: this section must only include enzymes comming from the gut microbiota as you are refering to that in the review.
Response 7: Thanks for the reminder, we make sure this section and title are accurate.
Comment 8: - line 110: "exogenously" this section is entitled enzymes from the microbiota and you are descriving the gut commensal microbiota, all the enzymes supplemented, independelty if they came or not from bacterial synthesis, must be moved to "regulation of external factors section - line 171" as they are supplementes and then must be considered external.
Response 8: Thank you for your insightful questions and reminders. We were not able to accurately convey the meaning of that section. We moved this part to ‘regulation of external factors’ section and we have rewritten this section (2.1.1 Enzymes coming from the microbiota) to make sure it's relevant to the topic.
Comment 9: - line 119: "feeding", same comment as previous. All the enzymes supplemented, independelty if they came or not from bacterial synthesis, must be moved to "regulation of external factors section - line 171" as they are supplementes and then must be considered external.
Response 9: Thank you for your suggestion. We have already fixed similar problems and moved the contents to ‘2.1.4 Enzymes coming from external factors’. Line 225-230.
Comment 10: - line 121: what do you mean by examining the rats eating...." this sentence needs to be improved from a scientif and english point of view.
Response 10: Thank you for your suggestion, and we reevaluated the reference and removed this citation.
Comment 11: - line128: this section must only include enzymes comming from the host as you are refering to that in the review.
Response 11: We make sure that this section corresponds to the content of the title (2.1.2 Enzymes coming from the host).
Comment 12: - line 132: "feed" if enzymes are fed, this need to be moved to "regulation of external factors section - line 171" as they are supplementes and then must be considered external.
Response 12: Thank you for your suggestion. We have moved these relevant contents to the ‘2.1.4 Enzymes coming from external factors’. Line 267-271.
Comment 13: - line 165: Bacteroides Thet... write in italics.
Response 13: We thank the Reviewer for his/her carefully reading. This part has been modified in the revised manuscript. Line 211.
Comment 14: - line 171: change title to Enzymes comming from ecternal factors or similar, to include all your observations not comming from the gut microbiota or the host.
Response 14: We changed the title as ‘Enzymes coming from external factors’. Line 216. Line 374.
Comment 15: - line 180: have in mind for this section all the comments from the last section (2.1).
Response 15: We sincerely thank the reviewers for their careful reading. Your detailed feedback has reinvigorated our research! We have revised the section strictly in accordance with your request to ensure that the content is consistent with the title (2.2. Non-nutritional functions of enzymes and their effects on microbiota).
Comment 16: - line 181: rewrite the introductory paragraph as done in line 100 to maintain homogeineity of the review.
Response 16: Thank you for your suggestion. We have modified the paragraph as you requested. Line 273-289.
Comment 17: - Table 1, lysozyme also comming from paneth cells no?
Response 17: Yes, you are correct. Lysozyme is indeed produced by Paneth cells in addition to other sources like saliva, tears, and egg whites. And we revised this part in the Table 1.
Comment 18: - line 623: summary must go to final summary and conclusions section
Response 18: Thank you for your suggestion. We moved this paragraph to the conclusions section.
Comment 19: - Table 2: for ACE2 change Host by the cell types secreting it
Response 19: Thank you for your suggestion. We added the ‘type II pneumocytes in the lungs’ in Table 2.
Comment 20: - Table 2: for Alkaline Phospatase include the cell types secreting it
Response 20: Thank you for your suggestion. We added the ‘Hepatocytes (liver)- Osteoblasts (bone)- Enterocytes (intestinal epithelium)’ in Table 2.
Comment 21: - Table 2: for DAO change Host by the cell types secreting it
Response 21: Thank you for your suggestion. We added the ‘Enterocytes of the small intestine’ in Table 2.
Comment 22: - Table 2: for ACE2 change Host by the cell types secreting it
Response 22: Thank you for your suggestion. We added the ‘type II pneumocytes in the lungs’ in Table 2.
Comment 23: - Table 2: for Oligoscharide degradins enzyme change Yeast by the yest specie secreting it
Response 23: Thank you for your suggestion. We added the ‘Saccharomyces cerevisiae’ in Table 2.
Comment 24: - line 987: equalize reference style
Response 24: Thank you for your suggestion. We corrected the format of this reference. Line 832.
Comments on the Quality of English Language
Comment 25: English from few parts must be improved to make the review undestandable. See previous section.
Response 25: Thank you for your recognition of the majority content of the manuscript, but there may be some errors in the manuscript, and we had revised it. Besides, the text was replaced with correct language and better descriptions. We really hope that the flow and language level have been substantially improved.